REGISTERED REPORT PROTOCOL

# Reproducibility of knee extensor and flexor contraction velocity in healthy men and women assessed using tensiomyography: A study protocol

**Georg Langen**[ID][1,2]*, **Christine Lohr**[3¤], **Olaf Ueberschär**[ID][4,5], **Michael Behringer**[1]

**1** Department of Sports Medicine and Performance Physiology, Goethe University Frankfurt, Frankfurt, Germany, **2** Department of Technical-Tactical Sports, Institute for Applied Training Science, Leipzig, Germany, **3** Independent Researcher, Germany, **4** Department of Engineering and Industrial Design, Magdeburg-Stendal University of Applied Sciences, Magdeburg, Germany, **5** Department of Biomechanics, Institute for Applied Training Science, Leipzig, Germany

¤ Current address: Osteopathiepraxis, Hamburg, Germany
* langen@sport.uni-frankfurt.de

## Abstract

Tensiomyography measures the radial displacement of a muscle during an electrically evoked twitch contraction. The rate of muscle displacement is increasingly reported to assess contractile properties. Several formulas currently exist to calculate the rate of displacement during the contraction phase of the maximal twitch response. However, information on the reproducibility of these formulas is scarce. Further, different rest intervals ranging from 10 s to 30 s are applied between consecutive stimuli during progressive electrical stimulation until the maximum twitch response. The effect of different rest intervals on the rate of displacement has not been investigated so far. The first aim of this study is to investigate the within and between-day reliability of the most frequently used formulas to calculate the rate of displacement. The second aim is to investigate the effect of changing the inter-stimulus interval on the rate of displacement. We will determine the rectus femoris and biceps femoris rate of displacement of twenty-four healthy subjects' dominant leg on two consecutive days. The maximum displacement curve will be determined two times within three minutes on the first day and a third time 24 h later. On day two, we will also apply three blocks of ten consecutive stimuli at a constant intensity of 50 mA. Inter-stimuli intervals will be 10 s, 20 s or 30 s in each block, respectively, and three minutes between blocks. The order of inter-stimulus intervals will be randomized. This study will allow a direct comparison between the five most frequently used formulas to calculate the rate of displacement in terms of their reproducibility. Our data will also inform on the effect of different inter-stimulus intervals on the rate of displacement. These results will provide helpful information on methodical considerations to determine the rate of displacement and may thus contribute to a standardized approach.

**Data Availability Statement:** All relevant data from this study will be made available upon study completion.

**Funding:** The authors received no specific funding for this work.

**Competing interests:** The authors have declared that no competing interests exist.

## Introduction

A muscle twitch is the contractile response to a single electrochemical signal of the nervous system or artificial electrical stimulation of the muscle. As such, a twitch provides information on muscle contractile properties and the functioning of the excitation-contraction coupling process. Tensiomyography (TMG) measures the radial displacement of a muscle belly during an electrically stimulated isometric twitch response. From the radial displacement curve, spatial and temporal parameters are derived. The two most frequently reported parameters are the maximum displacement (Dm) and the contraction time (Tc) [1]. Dm provides information on skeletal muscle stiffness, morphological and structural changes [2–4]. Tc refers to the time interval between 10% and 90% of Dm and is correlated to the proportion of slow-twitch fibres [5–7]. Therefore, a shorter Tc is commonly associated with a higher contraction velocity [8–11].

TMG is an involuntary method that does not require any effort from the subject. Therefore it is frequently used to assess muscular function following fatiguing exercise [12–16] or the effectiveness of different recovery strategies [17–20]. Muscular fatigue causes a slowing of muscle contraction velocity, reversing as the muscle recovers from fatigue [21–24]. Consequently, an increase in Tc of a fatigued muscle which is reversed as the muscle recovers is expected. However, previous studies found no change [14] or even a decrease of Tc [16, 25] after fatiguing exercise. Several authors pointed out that changes in Dm inevitably lead to changes in Tc [25, 26]. Accordingly, changes in Tc should not be interpreted independently of changes in Dm.

As a result, an increasing number of studies reports the rate of displacement (Vc), i.e. the slope of the radial displacement-time curve. Several different formulas exist to calculate Vc but no consensus exists on the most suitable formula. Further, there is a lack of studies investigating the reproducibility of these formulas, as only two studies investigated the reliability of Vc so far. Lohr et al. [27] investigated the within and between-day reliability of Vc in the lumbar erector spinae muscle of healthy females and males, using this formula: $Vc = 0.8{*}Dm / Tc$. Vc in this study showed an excellent relative (intraclass correlation coefficient ICC > 0.90) and absolute (coefficient of variation CV < 8%) reliability. De Paula Simola et al. [28] investigated the between-day reliability of the following two formulas to calculate Vc in twenty male sports students: $V_{10} = 0.1{*}Dm / t_{10\%}$ and $V_{90} = 0.9{*}Dm / t_{90\%}$ [28]. Absolute reliability was higher in rectus femoris (RF) and biceps femoris (BF) (CV < 10%) compared to gastrocnemius lateralis (CV = 12.3% and CV = 11.3% for $V_{10}$ and $V_{90}$, respectively) [28]. Both formulas showed excellent relative reliability in all three muscles (ICC > 0.90) [28]. In addition to the formulas used in the two studies mentioned, two other formulas to calculate Vc have been frequently reported: $Vc = Dm / (Td + Tc)$ [29] and $Vc_{rn} = 0.8 / Tc$ [30]. The reproducibility of Vc calculated by these two formulas is still unclear. Therefore, information on the reproducibility of these two formulas is needed. Further, a direct comparison of all five formulas for calculating Vc in terms of their reproducibility within the same sample would help reach a consensus on a standardized approach.

Differences exist not only regarding the formulas used to calculate Vc, but also regarding the electrical stimulation procedure to determine the radial displacement curve from which Vc is calculated. In a systematic review about the reliability of TMG, Martín-Rodríguez et al. [1] found that previous studies mainly used an inter-stimulus interval (ISI) of 10 s or 15 s to avoid fatigue or post-tetanic potentiation. But they also stated a lack of studies investigating the optimal rest interval between consecutive stimuli [1]. To our knowledge, only two studies investigated the effect of different ISI on tensiomyographic parameters so far. One study investigated the effect of different ISI (30 s, 10 s, 5 s) on Dm and Tc of the RF muscle during ten consecutive stimuli at a constant intensity (50 mA) [31]. The results showed that Tc was not significantly affected by different ISI, whereas Dm was significantly greater using an ISI of 10 s than 30 s

[31]. Notably, Tc decreased significantly during ten consecutive stimuli, irrespective of the ISI, whereas Dm was not affected [31]. However, Latella et al. [32] found no significant effect of ISI (10 s, 20 s) on Dm or Tc of the biceps brachii muscle. Consequently, they concluded that an ISI of 10 s is sufficient during five consecutive stimuli [32]. However, both of the studies mentioned above did not report Vc, so the effect of ISI during repeated stimulation on Vc is still unclear.

Therefore, the first aim of this study will be to investigate the within and between-day reliability of five different formulas to calculate Vc. The second aim will be to investigate the effect of different ISI during repeated stimulation on Vc, assessed by the example of M. biceps femoris and M. rectus femoris. We hypothesize that Vc will be affected by changing ISI during ten repeated stimuli at a constant intensity of 50 mA.

## Materials and methods

### Study design

This study will be a single group reliability study with repeated measurements within two consecutive days. The design of this study follows the recommendations for conducting and reporting diagnostic reliability studies provided in the quality appraisal of reliability studies (QAREL) checklist [33] and the guidelines for sex and gender equity in research (SAGER) [34].

### Ethical approval, registration and data availability

This study will be conducted in accordance with the Declaration of Helsinki [35]. The Ethics Committee of the Institute of Applied Training Sciences has approved this study (reference number: ER_2021.29.03_2). Eligible participants will be informed about all relevant aspects of this study and asked for their written consent before their enrolment. Upon in principle acceptance from the journal and before starting the recruitment and data collection, this study protocol will be preregistered. The preregistration including the accepted study protocol and supporting information will be openly available on the Open Science Framework platform at https://osf.io/4du2j/?view_only=ee7a4afb30514cb9a2e15381282dcec8. Once we finished the data collection and analysis, our data and the analysis code will be also openly available on this platform.

### Participants and setting

This study will be conducted at the Institute for Applied Training Science. We will include 24 women and men in this study, who will be required to meet the following inclusion criteria: healthy, aged between 18–40 years, physically active for a minimum of three times per week. In addition, we will describe the subjects' physical activity level based on data collected using the short form of the International Physical Activity Questionnaire (IPAC-SF) [36]. Exclusion criteria will be the following: pregnancy, history of neuromuscular or musculoskeletal disorders, pain or injury in the lower limbs during the last six months, previous surgical treatment to the lower limbs, practising a specific sport on a professional level, taking prescribed medication, nontolerance or any contraindication to electrical stimulation using self-adhesive electrodes, and wearing an implantable medical device. Participants will be recruited from scientific staff and students from the Institute for Applied Training Science and the Sport Sciences Department of the University of Leipzig. To this end, we will promote the study through email distribution lists, flyers and word of mouth. Potential participants will receive an informal invitation.

## Experimental approach

We will determine the absolute and relative within- and between-day reliability of the five most frequently used formulas to calculate the rate of displacement of the RF and BF assessed via TMG. The RF and BF were selected as they are the most frequently investigated muscles within studies reporting Vc. We will also determine the effect of three different rest intervals between ten consecutive stimuli at a constant intensity on the rate of displacement. The number of stimuli is consistent with a previous study which showed that Tc decreased significantly with ten stimuli [31], and which served as the reference for our sample size calculation. Fig 1 schematically illustrates the study flow.

A single rater with more than three years of experience performing TMG measurements will perform all tensiomyographic measurements across two consecutive days. The individual maximum radial displacement curve of both the RF and BF will be determined two times on the first day (M1, M2), separated by a three-minute pause, and a third time on the second day (M3), 24 h after the first measurement. On the second day and three minutes after the third measurement, we will apply three blocks of ten consecutive stimuli each. The rest intervals between consecutive stimuli in each block will be 10 s, 20 s and 30 s, respectively. The rest period between block 1, block 2 and block 3 will be three minutes, respectively. We will randomize the order of the three blocks with the different ISI for each participant.

On the first day, we will familiarize all participants with the electrical stimulation procedure by applying two stimuli with a duration of 1 ms each at 20 mA and 30 mA to the RF, followed by a three-minute rest before starting the actual measurement [31]. All measurements will be taken at the same time of the day, and a constant room temperature of 21° C. Participants will be asked to refrain from caffeine intake for 2 h preceding all measurements [37] and to avoid alcohol consumption and fatiguing exercise for 24 h before the start and during the trial to counteract possible confounding. Further, participants will be asked to record their total dietary intake during the 24 h before the first appointment and replicate their intake during the 24 h before the second visit [26] (S1 Appendix).

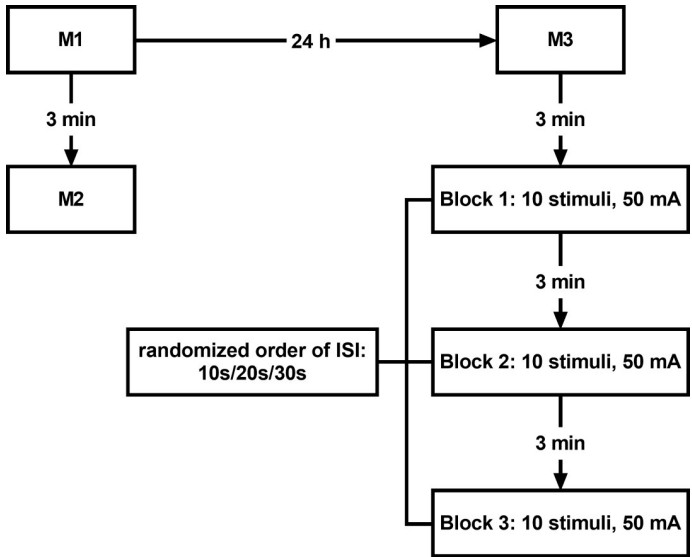

**Fig 1. Schematic illustration of the study flow.** M1, M2, M3: Measurement time points 1,2,3, respectively, mA: Milliampere, ISI: Inter-stimuli interval.

### Randomization and blinding

We will use an online program [38] and a block randomization procedure with four blocks of six unique numbers [39] to determine the order of ISI during repeated stimulations for each participant. To blind the rater from previous measurements, we will choose the settings of the TMG measurement software as not to display prior measurements.

### Sample size justification

Two different calculations were made to justify the sample size for this study. First, a precision-based calculation for the ICC of Vc was based on previous reliability studies [27, 28]. For this calculation, we used RStudio, Version 1.3.959 and the "Nest"-function from the "ICC" package version 2.3.0 [40]. Assuming the lowest ICC reported of 0.92 [28], two measurements per subject and an alpha level of 0.05, we will need 18 subjects to achieve the desired precision with a confidence interval of 0.15.

Second, to estimate how many subjects we will need to detect an effect of changing the interstimulus-interval on Vc, we used the raw data provided by Wilson et al. [31]. From those data, we calculated $Vc_{10\text{-}90\%} = 0.8\ Dm\ /\ Tc$ and $Vc_{norm} = 0.8\ /\ Tc$ (S1 Table). As only Dm and Tc were included in those raw data, we could not apply the other three formulas. We performed a two-way (ISI x measure) repeated measures analysis of variance (ANOVA) to estimate the main effect of inter-stimulus interval on $Vc_{10\text{-}90\%}$ and $Vc_{norm}$. We obtained effect sizes using RStudio and the "eta2_to_f"-function from the "effectsize" package [41] and calculated the sample size using the "wp.rmanova"-function from the "WebPower"-package [42]. For the sample size calculation, we assumed three groups, ten measurements per subject, an effect size $f = 3.1$ ($Vc_{10\text{-}90\%}$) and $f = 1.6$ ($Vc_{norm}$), a nonsphericity correction coefficient of 0.7 ($Vc_{10\text{-}90\%}$) and 0.6 ($Vc_{norm}$), an alpha level of 0.01, a power of 0.95 and a between effect, a total of 9 ($Vc_{10\text{-}90\%}$) subjects and 18 ($Vc_{norm}$) subjects were needed. We confirmed the estimated sample size for the ANOVA by repeating the calculations for both $Vc_{10\text{-}90\%}$ and $Vc_{norm}$ using g*power v3.1.9.2 [43]. Based on the results of both calculations for the ICC and the ANOVA and accounting for potential dropouts, we intend to recruit 24 subjects.

### Experimental set-up and procedures

To perform tensiomyographic measurements, we will use a TMG-S1 electrical stimulator (TMG-BMC d.o.o., Ljubljana, Slovenia), a GD30 displacement sensor (Panoptik d.o.o., Ljubljana, Slovenia) and two squared self-adhesive electrodes (50x50 mm, Axion GmbH, Leonberg, Germany). The signal of the displacement sensor will be recorded using the TMG Software v3.6 (TMG-BMC d.o.o., Ljubljana, Slovenia).

All measurements will be performed on the RF and BF of the dominant leg. The dominant leg will be the leg that subjects reportedly would use to shoot a ball on a target [44]. The RF measurements will be performed with subjects lying in a supine position, with their arms rested aside and the knee supported at an angle of approximately 120° by a triangular foam pad provided by the manufacturer [31]. The BF measurements will be performed with subjects lying in a prone position, with their arms rested aside and the ankle supported by a semicircular foam pad, creating a knee angle of approximately 175° flexion [28]. For both muscles, 180° corresponds to full knee extension, respectively.

The skin in the area of measurement will be cleaned with alcohol-soaked gauze before the positioning of the sensor and the electrodes. The position of the sensor will be determined in three steps: First, the midpoint on a line between the superior border of the patella and the anterior superior iliac spine for the RF and the BF, the midpoint on a line between the fibula head and ischial tuberosity will be determined, according to [45]. Second, the thickest part of

the muscle belly in the area of the point determined during the first step will be identified by inspection and palpation during a voluntary contraction. Third, if necessary, the sensor position will be adjusted at the beginning of the progressive stimulation to obtain the highest radial displacement [26, 27]. Once the position of the sensor is identified, we will place the electrodes at a distance of 7 cm from each other, measured between the facing edges of the two electrodes, and a distance of 3.5 cm between the sensor and the proximal and distal electrodes, respectively [46]. The position of the sensor and electrodes will be marked using a dermatological pen to ensure consistent positioning during this study.

The electrical stimulation will consist of single, monophasic, square wave stimuli with a duration of 1 ms each to elicit single isometric twitches. The stimulation will start at an initial stimulation amplitude of 30 mA, and will be then increased in steps of 10 mA. During M1, M2 and M3, we will apply progressive electrical stimuli with rest-intervals of 15 s between consecutive stimuli to obtain the individual maximal displacement. The stimulation amplitude will be increased until there is no further increase of the maximum radial displacement or until the stimulator's maximum output (110 mA) is reached. During the three blocks of repeated stimulation at 50 mA on the second day, the rest intervals between consecutive stimuli will be 10 s, 20 s and 30 s, respectively.

From M1, M2 and M3, the two displacement curves with the highest radial displacement of each measurement will be averaged respectively and used for further analysis. From block 1, block 2 and block 3, we will use every single displacement curve for further analysis.

## Rate of displacement calculation formulas

Fig 2 illustrates three generic tensiomyographic parameters and the five different approaches to calculate Vc from the displacement curve. The TMG software will automatically calculate the following parameters (Fig 2A): maximum radial displacement (Dm, mm), delay time, which refers to the time interval between the stimulus and 10% of Dm (Td, ms) and contraction time, which refers to the time interval between 10% and 90% of Dm (Tc, ms).

From these data, we will calculate (1) the mean rate of displacement during the time interval between the electrical stimulus until 10% of Dm as $Vc_{0-10\%} = 0.1 * Dm / Td$ (mm/s) [17] (Fig 2B), (2) the mean rate of displacement during the time interval between the electrical stimulus until 90% of Dm as $Vc_{0-90\%} = 0.9 * Dm / (Td + Tc)$ (mm/s) [25] (Fig 2B), (3) the mean rate of displacement during 10% and 90% of Dm as $Vc_{10-90\%} = 0.8 * Dm / Tc$ (mm/s) [17] (Fig 2B), (4) the normalized rate of displacement by dividing the mean rate of displacement during 10% and 90% of Dm by Dm as $Vc_{norm} = 0.8 / Tc$ (1/s) [30] and (5) the ratio of Dm and the time interval from the stimulus until 90% of Dm as $Vc_{Dm/t90\%} = Dm / (Td + Tc)$ (mm/s) [29] (Fig 2A).

## Statistical analysis

We will report $Vc_{0-10\%}$, $Vc_{0-90\%}$, $Vc_{10-90\%}$, $Vc_{norm}$ and $Vc_{Dm/t90\%}$ for both the RF and BF descriptively as mean (M) and standard deviation (SD). All data will be checked for outliers via visual inspection of boxplots. Outliers will be defined as data points outside of three times the interquartile range. The normal distribution of all variables will be tested using the Shapiro-Wilk Test. To assess a systematic bias between M1-M2 and M1-M3, we will perform a two-tailed paired t-test, respectively [47].

For the assessment of relative reliability, we will calculate intra-class correlation coefficients (ICC, mean rating (k = 2), absolute agreement, two-way mixed-effects model) with 95% confidence intervals (CI) [48]. ICC scores lower than 0.5, between 0.5 and 0.75, between 0.75 and 0.9 or greater than 0.9 will be interpreted as poor, moderate, good or excellent reliability [48].

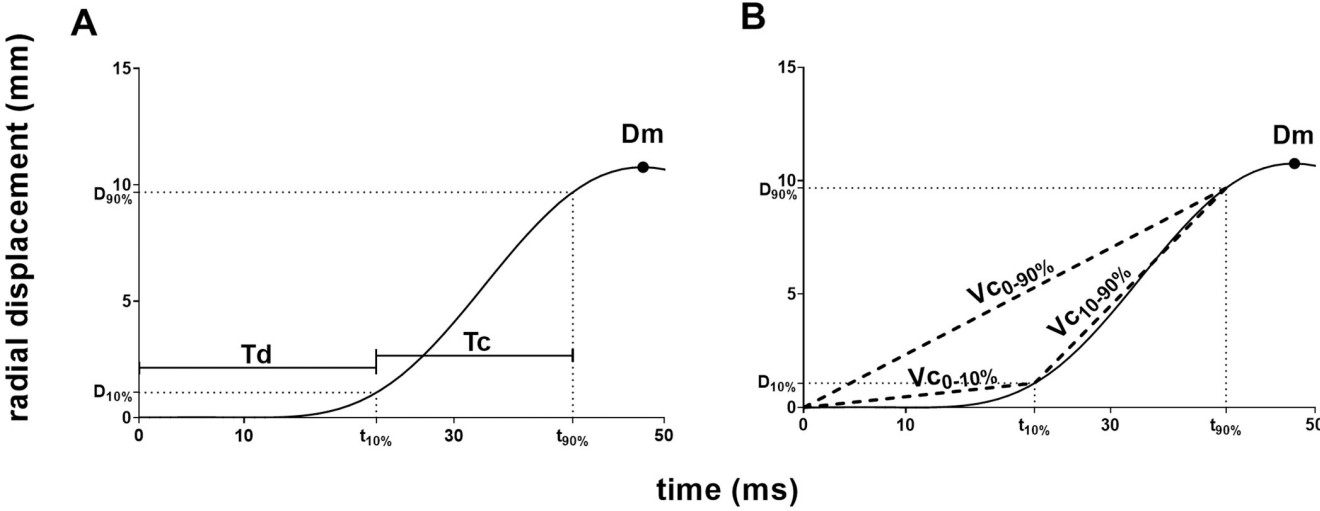

**Fig 2. Typical displacement curve during the twitch contraction phase until maximum displacement and tensiomyographic parameters.** (A) Three generic tensiomyographic parameters: maximum radial displacement (Dm), Delay time (Td), Contraction time (Tc). (B) Mean rate of displacement from electrical stimulus until 10% of Dm ($Vc_{0-10\%}$), mean rate of displacement from electrical stimulus until 90% of Dm ($Vc_{0-90\%}$) and mean rate of displacement between 10% and 90% of Dm ($Vc_{10-90\%}$).

Absolute reliability will be quantified by calculating the standard error of measurement (SEM and SEM%), the minimal detectable change (MDC and MDC%) [49] and the CV [47], including the respective 95% CI's. The SEM indicates the estimation precision of a measure and will be calculated as $SEM = SD * \sqrt{1-ICC}$ [47]. The SEM% allows a comparison of different Vc calculations as it is unitless and will be calculated as $SEM\% = (SEM / M) * 100$ [50]. The MDC presents the smallest difference between two measures that can be differentiated from measurement error and hence may be considered real [49]. Following Weir et al. [49], we will calculate the MDC as $MDC = SEM * 1.96 * \sqrt{2}$. The MDC% will be calculated as $MDC\% = (MDC/M) * 100$ [50] to allow a comparison between different calculations of Vc independently of units. The CV will be calculated as $CV = SD / M * 100$ [47], whereby SD refers to the standard deviation and M to the mean of individual differences between M1 and M2 or M1 and M3. A CV > 10% will be interpreted as insufficient reliability with reference to previous studies investigating the reliability of tensiomyographic parameters [27, 51, 52].

To assess the effects of different ISI on Vc during repeated stimulation, we will employ a within-subject repeated-measures ANOVA. The two independent variables will be ISI (10 s, 20 s, 30 s) and stimulus number (stimulus 1 to stimulus 10). The dependent variables will be $Vc_{0-10\%}$, $Vc_{0-90\%}$, $Vc_{10-90\%}$, $Vc_{norm}$ and $Vc_{Dm/t90\%}$ for both the RF and BF, respectively. Pairwise comparisons with Bonferroni correction will applied for significant main effects. The effect size Cohen's d will be calculated as $d = M_{diff} / \sqrt{SD_1 + SD_2} / 2$ [53] for pairwise comparisons to allow comparisons between effects of different ISI and between consecutive stimuli. Thresholds for small, moderate, large, very large or extremely large effects will be 0.2, 0.6, 1.2, 2.0 and 4.0 [54].

Statistical significance will be set at alpha ≤ 0.05. Missing data will be handled by listwise deletion of cases with missing data [55]. All statistical analyses will be performed using R studio and respective packages. Upon completion of the data collection and analysis, the raw data disaggregated by sex and the analysis code will be openly available at https://osf.io/4du2j/?view_only=ee7a4afb30514cb9a2e15381282dcec8.

## Discussion

This study will compare the five most frequently used formulas to calculate Vc regarding their within and between-day reliability, assessed by the example of M. biceps femoris and M. rectus femoris. This study will also investigate the effect of different ISI during repeated stimulation on Vc of the same muscles.

To our knowledge, only two studies have investigated the reproducibility of Vc so far. Lohr et al. [27] investigated the within and between-day reliability of $Vc_{10-90\%}$ in the lumbar erector spinae muscle of healthy females and males. $Vc_{10-90\%}$ in this study showed an excellent relative (ICC > 0.90) and absolute (CV < 8%) reliability. De Paula Simola et a. [28] investigated the between-day reliability of $Vc_{10\%}$ and $Vc_{90\%}$ in the rectus femoris, biceps femoris, and gastrocnemius lateralis of male sports students. $Vc_{10\%}$ and $Vc_{90\%}$ both showed excellent relative reliability in all three muscles (ICC > 0.90) [28]. Absolute reliability was higher in rectus femoris and biceps femoris (CV < 10%) compared to gastrocnemius lateralis (CV = 12.3% and CV = 11.3% for $V_{10}$ and $V_{90}$, respectively) [28]. Our study will add to these results by contributing information on the within and between-day reliability of the five most frequently used formulas to calculate Vc. Thereby our results will allow a direct comparison between these formulas in terms of their reproducibility.

Further, information on the optimal rest interval between consecutive stimuli during progressive electrical stimulation is lacking. Again, only two studies have investigated the effect of different ISI on tensiomyographic parameters [31, 32]. One study investigated the effect of different ISI (30 s, 10 s, 5 s) on Dm and Tc of the rectus femoris muscle during ten consecutive stimuli [31]. The results showed that Tc was not significantly affected by different ISI, whereas Dm was significantly greater using an ISI of 10 s than 30 s [31]. Notably, Tc decreased significantly during ten consecutive stimuli, irrespective of the ISI, whereas Dm was not affected [31]. These findings are relevant for assessing the muscle contraction velocity using Vc. An increase in Vc can be expected, if Dm increases while Tc remains unchanged as a higher displacement amplitude is achieved within the same time. Similarly, an increase in Vc can be also expected, if Tc shortens while Dm remains unchanged, as the same displacement amplitude is achieved within a shorter time. According to Wilson et al. [31], these changes in Dm and Tc during repeated stimulation may be explained by a potentiation effect. Indeed, repeated low-frequency electrical stimulation can induce post-tetanic potentiation [56], which leads to increased peak force, decreased time to peak force and increased rate of force development during an electrically evoked twitch [57]. Thus, it is conceivable that such a potentiation effect could also lead to an increase in Vc.

In contrast, Latella et al. [32] found no significant effect of ISI (10 s, 20 s) on Dm or Tc of the biceps brachii muscle, measured across three different joint angles and two days. Consequently, they concluded that an ISI of 10 s is sufficient during five consecutive stimuli to avoid the effects of fatigue or potentiation. However, both studies did not report Vc, so the effect of ISI during repeated stimulation on Vc is still unclear. Therefore, our results will contribute relevant information on methical considerations for the determination of Vc using TMG.

A certain limitation of our study is that we will only assess two different muscles of the lower extremities. In the future, the reproducibility of Vc and the effect of changing ISI on Vc should therefore be investigated in a broad range of muscles, including muscles of the upper extremities. Another limitation of our study is that we will include only healthy and physically active women and men between 18 and 40. The effect of different ISI on Vc might differ in older subjects or athletes due to age- and training-related shifts in the muscle fibre spectrum [11, 58, 59].

## Supporting information

**S1 Appendix. 24 h dietary intake protocol template.**
(PDF)

**S1 Table. Calculated rate of displacement data used for sample size estimation.**
(XLSX)

## Author Contributions

**Conceptualization:** Georg Langen.

**Formal analysis:** Georg Langen, Christine Lohr.

**Investigation:** Georg Langen.

**Methodology:** Georg Langen, Christine Lohr, Olaf Ueberschär, Michael Behringer.

**Supervision:** Michael Behringer.

**Writing – original draft:** Georg Langen, Christine Lohr.

**Writing – review & editing:** Georg Langen, Christine Lohr, Olaf Ueberschär, Michael Behringer.

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
