## [Decision Letter · Decision Letter 0]

20 Nov 2021

PONE-D-21-24120Reproducibility of knee extensor and flexor contraction velocity in healthy men and women assessed using tensiomyography: A study protocolPLOS ONE

Dear Dr. Langen,

Thank you for submitting your manuscript to PLOS ONE. After careful consideration, we feel that it has merit but does not fully meet PLOS ONE’s publication criteria as it currently stands. Therefore, we invite you to submit a revised version of the manuscript that addresses the points raised during the review process.

We look forward to receiving your revised manuscript.

Kind regards,

Emiliano Cè

Academic Editor

PLOS ONE

Journal Requirements:

Reviewers' comments:

Reviewer's Responses to Questions

**Comments to the Author**

1. Does the manuscript provide a valid rationale for the proposed study, with clearly identified and justified research questions?

Reviewer #1: Yes

Reviewer #2: Yes

2. Is the protocol technically sound and planned in a manner that will lead to a meaningful outcome and allow testing the stated hypotheses?

Reviewer #1: Yes

Reviewer #2: Yes

3. Is the methodology feasible and described in sufficient detail to allow the work to be replicable?

Reviewer #1: Yes

Reviewer #2: Yes

4. Have the authors described where all data underlying the findings will be made available when the study is complete?

Reviewer #1: Yes

Reviewer #2: Yes

5. Is the manuscript presented in an intelligible fashion and written in standard English?

Reviewer #1: Yes

Reviewer #2: Yes

6. Review Comments to the Author

You may also provide optional suggestions and comments to authors that they might find helpful in planning their study.

Reviewer #1: General comment

Thank you for your thesis. Your research is very useful in establishing the reliability of TMG's methods. This study will be to investigate the within and between-day reliability of five different formulas to calculate Vc. The second aim will be to investigate the effect of different ISI during repeated stimulation on Vc, assessed by the example of M. biceps femoris and M. rectus femoris. However, there are some concerns that you should modify. Please refer following comments.

Line 107

Please explain the reason of selection in muscle (RF and BF) for your study.

Line 127

Please describe the location (setting) of your planned research.

Line 134

The description of IPAC-SF should be combined with the inclusion criteria of line129.

Line 141

When RF and BF are first mentioned, please use the formal names.

Line 142

Why will you perform your study with ten stimuli? Please add the reason the number of the stimuli in method.

Line 195

Please consider whether you will measure the thickness of the skin or subcutaneous tissue at the TMG measurement point. This may affect the measurement.

Reviewer #2: General comments:

This study protocol aims to investigate the reliability of Tensiomyography derived velocity calculations, the effect of inter-stimulus interval on velocity of contraction in the rectus femoris and biceps femoris. Overall, this study protocol is well written, with robust methodological considerations. This study protocol will benefit both practitioners and researchers to standardise which velocity of contraction equation, and inter-stimulus interval to use. This study could benefit from using normalised response speed (Vrn), which is a much more manageable and easier to interpret metric.

Specific comments:

Rewording a few sentences and phrases might make it easier for the flow of the manuscript.

Introduction:

Line 52 – electrochemical signal of the nervous system, and/or artificial electrical stimulation of the muscles (e.g., electrical stimulator).

Line 58 – consider adding the following sentence instead “skeletal muscle stiffness, morphological, and structural changes”

Line 71 – wouldn’t the line of slope be what time of contraction (Tc) measures? Please explain further.

Line 73 – add “exists” after consensus

Line 84 – perhaps providing the formulas of the other two methods of Vc calculation can be beneficial to the readers.

Line 218 – some authors have used inter-electrode distances between 5-10 cm, could you explain why you chose 7cm in particular? What are its benefits over other inter-electrode distances?

Methodology:

Perhaps consider also adding normalised response speed (Vrn), as it provides a value that is much easier to interpret and manageable.

Discussion:

Line 319-320 – this line is a bit confusing, did you mean “an increase in Tc can be expected if Dm increases, even while Vc remains unchanged”?

7. PLOS authors have the option to publish the peer review history of their article (what does this mean?). If published, this will include your full peer review and any attached files.

Reviewer #1: No

Reviewer #2: **Yes: **Angus Hunter

---

## [Author Response · Author response to Decision Letter 0]

5 Dec 2021

Reviewer#1: General comment

Thank you for your thesis. Your research is very useful in establishing the reliability of TMG's methods. This study will be to investigate the within and between-day reliability of five different formulas to calculate Vc. The second aim will be to investigate the effect of different ISI during repeated stimulation on Vc, assessed by the example of M. biceps femoris and M. rectus femoris. However, there are some concerns that you should modify. Please refer following comments.

Thank you for the positive and constructive feedback. We have added our responses below the respective comments and marked the corresponding changes in the text.

Line 107

Please explain the reason of selection in muscle (RF and BF) for your study.

We selected the RF and BF because they are the most frequently investigated muscles within studies reporting Vc. This information comes from the results of a scoping review which we plan to submit within the next few weeks. This reviews aim is to provide an overview of currently applied methodical concepts to determine Vc. The preregistration is openly available here: https://doi.org/10.17605/OSF.IO/ZKQMV

We also added the following information in line 147:

"The RF and BF were selected as they are the most frequently investigated muscles within studies reporting Vc."

Line 127 

Please describe the location (setting) of your planned research.

The study will be conducted at the Institute for Applied Training Science in Leipzig, Germany. We also added the following information:

Line 130: "This study will be conducted at the Institute for Applied Training Science."

Line 141: "The study will be promoted via email distribution lists, flyers and word of mouth. Potential participants will be invited via informal invitation."

 

Line 134

The description of IPAC-SF should be combined with the inclusion criteria of line129.

Thank you for the advice. We moved the description of IPAC-SF up to line 133 so that it follows directly the inclusion criteria.

Line 141 

When RF and BF are first mentioned, please use the formal names.

Thank you for pointing this out. However, RF and BF are first mentioned in the introduction section in line 82 where the corresponding abbreviations are also introduced. 

Line 142 

Why will you perform your study with ten stimuli? Please add the reason the number of the stimuli in method.

We added the following information in line 149:

"The number of stimuli is consistent with a previous study which showed that Tc decreased significantly with ten stimuli [31], and which served as the reference for our sample size calculation."

Line 195

Please consider whether you will measure the thickness of the skin or subcutaneous tissue at the TMG measurement point. This may affect the measurement.

Thank you for this consideration. To our knowledge, there are conflicting findings on the influence of the skinfold thickness at the measurement point on TMG-parameters. In line with your suggestion, Calvo-Lobo et al. (10.1590/1806-9282.64.06.549) found a significant correlation between Dm and both the thickness of the skin and subcutaneous fat layer of the erector spinae muscle. In contrast, Diez-Vega et al. 2013 (10.1136/bjsports-2013-092558.61) found no significant correlation between Dm of the rectus femoris muscle and skinfold thickness. Similarly, de Paula Simola et al. 2015 (10.1519/JSC.0000000000000768) found no significant correlation between any of the TMG parameters of the rectus femoris muscle and the skinfold thickness at the measurement point. In view of these previous results, the influence of the skinfold thickness at the measuring point on the measurement results appears not yet clearly understood. However, we suggest that the skinfold thickness is not of crucial importance for determining the reproducibility of the different Vc formulas, as a change in skinfold thicknesses within the time period of this investigation is not to be expected.

 

Reviewer #2: General comments:

This study protocol aims to investigate the reliability of Tensiomyography derived velocity calculations, the effect of inter-stimulus interval on velocity of contraction in the rectus femoris and biceps femoris. Overall, this study protocol is well written, with robust methodological considerations. This study protocol will benefit both practitioners and researchers to standardise which velocity of contraction equation, and inter-stimulus interval to use. This study could benefit from using normalised response speed (Vrn), which is a much more manageable and easier to interpret metric.

Specific comments:

Rewording a few sentences and phrases might make it easier for the flow of the manuscript.

Dear Prof. Hunter, thank you very much for your positive feedback and helpful comments on our manuscript. We have added our responses to your comments and marked the respective changes in the text.

Introduction:

Line 52 – electrochemical signal of the nervous system, and/or artificial electrical stimulation of the muscles (e.g., electrical stimulator).

We added the following phrase in line 53:

“… or artificial electrical stimulation of the muscle.”

Line 58 – consider adding the following sentence instead “skeletal muscle stiffness, morphological, and structural changes”

Thank you for the advice. We have adopted the suggested wording in line 59.

Line 71 – wouldn’t the line of slope be what time of contraction (Tc) measures? Please explain further.

Vc corresponds mathematically to the slope of the displacement-time curve within a defined time interval, calculated as the quotient of the respective displacement and the time needed to achieve it. Accordingly, Vc indicates the velocity or rate of muscle belly displacement in this respective time interval. On the other hand, Tc corresponds the time it takes for the muscle belly displacement to increase from 10% to 90% of Dm. 

Perhaps our expression in English is misleading at this point. To make our wording more precise, we added the word “time” in line 73. 

However, we would appreciate any suggestions if there is a more accurate expression of “the slope of the displacement-time curve”.

Line 73 – add “exists” after consensus

Thank you for pointing this out. We added the word in line 74.

Line 84 – perhaps providing the formulas of the other two methods of Vc calculation can be beneficial to the readers.

Thank you for the good advice, we added the formulas in line 86.

Line 218 – some authors have used inter-electrode distances between 5-10 cm, could you explain why you chose 7cm in particular? What are its benefits over other inter-electrode distances?

We decided for an IED of 7 cm based the results of Wilson et al 2018 (10.1371/journal.pone.0191965) and Wilson et al 2019 (10.1088/1361-6579/ab1cef).

Both studies investigated the effect of changing the IED on contractile properties assessed via TMG. Wilson et al. 2018 investigated the RF muscle and found an increase in Dm when IED was increased from 5 cm to 7 cm, but no further increase when IED was increased to 9 cm or 11 cm. Wilson et al. 2019 investigated the BF muscle and found that Dm was higher with an IED of 7 cm compared to 6 cm, 5, cm or 4 cm. 

Wilson et al. 2019 also argued that if the IED is too large, the maximum stimulator output may be reached before the maximum Dm is obtained. This is in line with the findings of a study by Petrofsky et al. 2008 (10.1007/s00421-008-0700-3), who investigated the influence of the IED among other stimulation parameters on the distribution of the electrical current on the skin and the depth of its penetration into the muscle. Increasing the length of the IED from 10cm to 15cm or 20cm resulted in a more even, wider distribution of the current on the skin but a reduced penetration depth at the same time. The penetration depth of the current is an important factor, as the distribution of fiber types may differ at the muscle surface and deeper in the muscle tissue (Dahmane et al. 2005, 10.1016/j.jbiomech.2004.10.020). 

Consequently, an optimal IED would provide a wide distribution at the surface and at the same time an adequate penetration depth into the muscle, ensuring that as many motor units as possible are activated by the electric stimulus. Given the paucity of studies that have investigated the influence of the IED on TMG parameters, it can certainly be questioned whether an IED of 7 cm is optimal for TMG measurements of both BF and RF muscle. However, according to the current available evidence regarding IED configurations for these muscles, we believe that 7 cm is a reasonable choice.

 

Methodology:

Perhaps consider also adding normalised response speed (Vrn), as it provides a value that is much easier to interpret and manageable.

Thank you for your suggestion. Do you mean the Vrn calculated by the formula first suggested by Valencic et al. 1997? If so, then this would correspond the fourth formula we mentioned in line 261: Vcnorm = 0.8*Dm / Tc*Dm = 0.8 / Tc.

Discussion:

Line 319-320 – this line is a bit confusing, did you mean “an increase in Tc can be expected if Dm increases, even while Vc remains unchanged”?

Thank you for pointing this out. We want to express that both an increase in Dm while Tc remains unchanged and similarly a decrease in Tc while Dm remains unchanged will result in an increase in Vc. To hopefully make this point easier to understand, we changed this wording as follows (line 327):

“An increase in Vc can be expected, if Dm increases while Tc remains unchanged as a higher displacement amplitude is achieved within the same time. Similarly, an increase in Vc can be also expected, if Tc shortens while Dm remains unchanged, as the same displacement amplitude is achieved within a shorter time.”

---

## [Decision Letter · Decision Letter 1]

17 Dec 2021

Reproducibility of knee extensor and flexor contraction velocity in healthy men and women assessed using tensiomyography: A study protocol

PONE-D-21-24120R1

Dear Dr. Langen,

We’re pleased to inform you that your manuscript has been judged scientifically suitable for publication and will be formally accepted for publication once it meets all outstanding technical requirements.

Kind regards,

Emiliano Cè

Academic Editor

PLOS ONE

Additional Editor Comments (optional):

Reviewers' comments:

Reviewer's Responses to Questions

**Comments to the Author**

1. Does the manuscript provide a valid rationale for the proposed study, with clearly identified and justified research questions?

Reviewer #1: Yes

2. Is the protocol technically sound and planned in a manner that will lead to a meaningful outcome and allow testing the stated hypotheses?

Reviewer #1: Yes

3. Is the methodology feasible and described in sufficient detail to allow the work to be replicable?

Reviewer #1: Yes

4. Have the authors described where all data underlying the findings will be made available when the study is complete?

Reviewer #1: Yes

5. Is the manuscript presented in an intelligible fashion and written in standard English?

Reviewer #1: Yes

6. Review Comments to the Author

You may also provide optional suggestions and comments to authors that they might find helpful in planning their study.

Reviewer #1: Thanks for the correction. Your work will provide useful information for future readers of the TMG protocol. I look forward to your further research.

7. PLOS authors have the option to publish the peer review history of their article (what does this mean?). If published, this will include your full peer review and any attached files.

Reviewer #1: No

---

## [Editor Report · Acceptance letter]

22 Dec 2021

PONE-D-21-24120R1 

Reproducibility of knee extensor and flexor contraction velocity in healthy men and women assessed using tensiomyography: A study protocol 

Dear Dr. Langen:

I'm pleased to inform you that your manuscript has been deemed suitable for publication in PLOS ONE. Congratulations! Your manuscript is now with our production department. 

Kind regards, 

on behalf of

Professor Emiliano Cè 

Academic Editor

PLOS ONE